# The Energetic Costs of Uphill Locomotion in Trail Running: Physiological Consequences Due to Uphill Locomotion Pattern—A Feasibility Study

**DOI:** 10.3390/life12122070

**Published:** 2022-12-09

**Authors:** Paul Zimmermann, Nico Müller, Volker Schöffl, Benedikt Ehrlich, Othmar Moser, Isabelle Schöffl

**Affiliations:** 1Department of Cardiology, Klinikum Bamberg, 96049 Bamberg, Germany; 2Interdisciplinary Center of Sportsmedicine Bamberg, Klinikum Bamberg, 96049 Bamberg, Germany; 3Division of Exercise Physiology and Metabolism, Department of Sport Science, University of Bayreuth, 95447 Bayreuth, Germany; 4Department of Orthopedic and Trauma Surgery, Klinikum Bamberg, 96049 Bamberg, Germany; 5Department of Orthopedic and Trauma Surgery, Friedrich-Alexander University Erlangen-Nurnberg, 91054 Erlangen, Germany; 6School of Clinical and Applied Sciences, Leeds Beckett University, Leeds LS1 3HE, UK; 7Section of Wilderness Medicine, Department of Emergency Medicine, University of Colorado School of Medicine, Denver, CO 80045, USA; 8Interdisciplinary Metabolic Medicine Research Group, Division of Endocrinology and Diabetology, Medical University of Graz, 8036 Graz, Austria; 9Department of Pediatric Cardiology, Friedrich-Alexander-University Erlangen-Nurnberg, 91054 Erlangen, Germany

**Keywords:** trail running, cardiopulmonary exercise testing, uphill running, uphill walking, energetic demands, short trail running performance

## Abstract

The primary aim of our feasibility reporting was to define physiological differences in trail running (TR) athletes due to different uphill locomotion patterns, uphill running versus uphill walking. In this context, a feasibility analysis of TR athletes’ cardiopulmonary exercise testing (CPET) data, which were obtained in summer 2020 at the accompanying sports medicine performance center, was performed. Fourteen TR athletes (n = 14, male = 10, female = 4, age: 36.8 ± 8.0 years) were evaluated for specific physiological demands by outdoor CPET during a short uphill TR performance. The obtained data of the participating TR athletes were compared for anthropometric data, CPET parameters, such as $\dot{V}Emaximum$, $\dot{V}O_{2}maximum$, maximal breath frequency (BF_max_) and peak oxygen pulse as well as energetic demands, i.e., the energy cost of running (C_r_). All participating TR athletes showed excellent performance data, whereby across both different uphill locomotion strategies, significant differences were solely revealed for $\dot{V}Emaximum$ (*p* = 0.033) and time to reach mountain peak (*p* = 0.008). These results provide new insights and might contribute to a comprehensive understanding of cardiorespiratory consequences to short uphill locomotion strategy in TR athletes and might strengthen further scientific research in this field.

## 1. Introduction

Mountain endurance running, especially trail running (TR), has increased its popularity in the recent years [1]. The International Trail Running Association defined TR as a “pedestrian off road race in a natural environment (e.g., mountain) with minimal possible paved or asphalt road (<20% of the total duration race)” and TR profiles start with short distances (<42 km) and may be extended to ultralong distances (>100 km) [2]. The growing popularity of this sport has led to new scientific research fields in sports science referring to physiological consequences influencing an athlete’s performance determinants and factors influencing an individual athlete’s neuromuscular fatigue [1,3,4,5,6]. As TR represents a complex sport in terms of cardiorespiratory and biomechanical demands, individual athlete’s race performance prediction is challenging [7,8]. Therefore, evaluating clinical and sports performance, incremental running tests have been developed to elucidate maximal cardiorespiratory capabilities [9,10,11].

As previously reported, an athlete’s performance in level running depends on several influencing factors, such as the energy cost of running (Cr), the individual maximal oxygen uptake ($\dot{V}O_{2}maximum$), and the fraction of $\dot{V}O_{2}\mathrm{peak}$ that can be kept up during an athlete’s race performance [12].

In the recent decades, the growing popularity of TR has attracted interest in specific cardiorespiratory performance characterization of these athletes, including maximum oxygen uptake values ($\dot{V}O_{2}maximum)$ [13,14,15]. However, to date, few studies have evaluated the impact of a slope on specific maximal physiological values, such as $\dot{V}O_{2}maximum$, during incremental running tests [10]. The existing literature reveals contrasting results and shows the lack of consensus caused by different athlete’s population and/or testing protocols [10]. Therefore, on the one hand, previous research revealed similar results for $\dot{V}O_{2}maximum\;$in level and uphill running [1,11,16,17], while, on the other hand, contrary findings for the physiological demands during uphill sections in comparison to level conditions were elucidated previously in this scientific field [10,18,19,20]. Considering downhill running, previous research revealed that $\dot{V}O_{2}maximum$ can not be reached and that the $\dot{V}O_{2}maximum$ is estimated to be 16–18% lower than in level and uphill maximal incremental running tests [21]. Previous research in this specific scientific area by Schöffel et al., and Balducci et al., could not reveal significant changes in $\dot{V}O_{2}maximum\;$ with an increasing slope for TR athletes, but they did find a progressive increase in ventilation ($\dot{V}E)$ [1,22].

In contrast, other studies revealed a correlation between real-world uphill TR performance and certain parameters, such as running economy, maximal strength, local endurance assessment by fatigue index (FI), and the athlete’s characteristics, such as body fat percentage and athlete’s age [2,13,23]. The classical physiological variables of endurance running, such as $\dot{V}O_{2}maximum$ and/or percentage of $\dot{V}O_{2}maximum$ at ventilatory threshold (VT), did not allow meaningful prediction of short TR performance [2].

Next to the cardiorespiratory demands, the bioenergetic demands and biomechanical work for level and sloped surface running have been studied before [23,24,25,26,27]. As previously reported, the Cr in general depends on the characteristics of the terrain, on the incline as well as on the biomechanics of uphill locomotion and is independent of speed [12,27,28,29,30]. Several influencing factors on Cr might lead to interindividual athlete performance variability, such as higher resting metabolism and leg architecture, such as muscle strength of the plantar flexor muscle and triceps surae muscle, as well as a combination of eccentric and concentric actions forming the stretch-shortening muscle function resulting in elastic energy storage and reuse [31]. In fact, factors influencing the Cr of running are rather well identified in uphill versus level running, whereby additional influencing cardiorespiratory, metabolic, and biomechanical factors in highly trained or elite runners have been reported in previous research [32,33]. In this context, metabolic adaptations within the muscle, such as increased mitochondria and oxidative enzymes, and more efficient individual athlete’s mechanics requiring less energy are the main putative factors [32]. Considering that many factors are influencing Cr [31,32], it might be assumed that an individual runner’s characteristics contribute to different adaptive strategies to uphill locomotion with variable Cr levels in level and uphill running [1].

TR performance is multifaceted, and comparative studies on this scientific topic are rare up to now. While certain variable circumstances influence an athlete’s performance, the execution of comparable studies is hindered. In this context, variable determinants such as uphill locomotion strategy, the usage of poles to minimize energy expenditure, testing profiles and procedures as well as comparable homogeneous cohorts of athletes have to be taken into consideration when conducting a study and to better characterize short TR performance and the energy expenditure during uphill locomotion [2,22,34,35].

Hence, the aim of the presented feasibility study was to compare the cardiorespiratory variables in TR athletes in a short outdoor field-testing protocol focusing different uphill locomotion strategies, such as uphill running vs. uphill walking. It was hypothesized that there would be no differences in the TR CPET performance data between both analyzed uphill locomotion strategies. In detail, we evaluated the specific interindividual CPET performance data values due to the uphill locomotion strategy, such as $\dot{V}O_{2}maximum$, maximal ventilation ($\dot{V}Emaximum$), time to reach anaerobic VT, breath frequency (BF), peak oxygen pulse (peak O_2 pulse_), athlete’s blood lactate level, and athlete’s specific Cr data. Since the sport-specific cardiorespiratory demands in TR differ immensely from road running, the presented novel data allow a better understanding of the sport-specific physiological demands in short TR performance determined by the variable uphill locomotion strategy [22].

## 2. Materials and Methods

### 2.1. Study Design

This was a single-center feasibility study of outdoor field uphill CPET performance data from 14 participating TR athletes, which were obtained in 2020 during short outdoor field testing. After inclusion in the study, participating trail runners were assigned to ascending numbers and afterwards were allocated to the order in which the testing protocols (uphill running vs. uphill walking) were conducted in a cross-over randomized fashion with the software Research Randomizer 4.0 (Social Psychology Network^®^, Lancaster, PA, USA) (1:1) [36].

The obtained data of participating TR athletes (n = 14) were compared for the physiological consequences measured by CPET performance data and Cr parameters for short uphill running versus uphill walking locomotion.

### 2.2. Ethical Consideration

The study protocol (228_20 B) was approved by the local ethics committee of the University of Nurnberg-Erlangen. The research was conducted in conformity with the declaration of Helsinki and Good Clinical Practice [37]. Prior to any trial-related activities and data acquirement, our participating TR athletes gave their written informed consent and were informed about the study protocol and the following data measurements.

### 2.3. Participating Trail Running Athletes

Eligibility criteria included male or female TR athletes aged between 25 to 50 years with a body mass index (BMI) between 19 and 25 kg/m^2^. The characteristics of the included participants displayed an age of 36.8 ± 8.0 years, with a total height of 179 ± 8.4 cm, a body mass of 70.4 ± 10.0 kg, and BMI of 21.8 ± 1.8 kg/m^2^. Regular attendance to TR competitions with a minimum of 21 km distance was a precondition to take part in our study. Our participating athletes did not use poles during uphill locomotion testing.

During the study protocol, 2 participants of the initially recruited 16 athletes had to be withdrawn from the analysis due to technical problems during the outdoor field CPET data assessment. Hence, 14 data sets were included for analysis and are presented in the results section. Anthropometric data, training, and race performance parameters of participating athletes are displayed in Table 1 and Table 2.

By an individual questionnaire, each athlete was evaluated for the displayed training and race information in Table 2. In this context, our tapering period was defined by the following aspects: cutting back training up to 3–4 weeks prior to TR competition, reduced training volume (by roughly 30%), maintaining training frequency with upholding intensity (fewer repetitions, less miles), reduced high risk injury training sessions, and speed workouts. So, the tapering was estimated to display the right balance in our athletes between training volume, intensity, and frequency. These points were evaluated to discriminate between race and tapering period in our athletes. Referring to their best performance in an official 1000 m denivelation race, no slope, race length, or ground surfaces were defined or evaluated in our athletes. The purpose of the evaluation was to estimate roughly the individual athlete’s performance and the homogeneity of the athletic cohort; no specific evaluation of their race performance took place. Our participating athletes, who all took part in several TR competitions, were asked for their preferred locomotion strategy due to their race experience depending on individual TR profiles, ground surfaces, and TR distances. In this context, the athlete’s answers were based on their race assessment and personal experiences.

As a test site, the Wiesenttal mountain in the upper Franconian Switzerland was chosen. The topographical profile with a length of 375 m and a maximal incline of 29.3% (mean incline overall 22.3 ± 7%) represents an optimal testing area for short TR, and the mountain is part of an actual TR race course. Further information and the topographic profile of this TR race course are provided at https://www.outdooractive.com/de/route/trailrunning/fraenkische-schweiz/wiesenttal-trail-neideck-1000/105762273/#dm=1 (accessed on 23 October 2022). Individuals were excluded if they were enrolled in a different study, had a history of acute infection, and/or CPET testing was contraindicated. A medical investigator assessed inclusion and exclusion criteria before enrolment in the study. In this context, an infection-free interval of at least 4 weeks and no musculoskeletal injuries within the last 4 weeks were a precondition to be involved in the study.

### 2.4. Outdoor Uphill Field Measurements

Participants were instructed about all study-related procedures during the first visit. After assessing the anthropometric data and training and race performance data, the participating TR athletes were obligated to proceed with a warm-up for 10 min prior to uphill locomotion testing to be in comparable physiological readiness before the testing. A maximum of 5 min between the end of the warm-up and the beginning of testing was permitted. The outdoor field-testing data acquirement was scheduled during the weekend after having a small, not prescribed breakfast two hours prior to testing, and the athletes were asked to avoid intensive training units for two days prior to testing. Each athlete completed two trail runs at the test site according to the allocated testing protocol, one in an uphill running locomotion and the other in an uphill walking locomotion. During the outdoor uphill field CPET, participants received a chest belt for continuous heart rate (HR) monitoring via a Bluetooth smart HR sensor (chest belt Dual ANT+/Bluetooth smart; Kalenji, Decathlon^®^, S.A., Lille, France) for safety reasons and assessment of peak and post-exercise maximum HR levels (HR measured in beats per minute (bpm)). We acquired our CPET data during the outdoor sport-specific field testing by using the mobile field test spiroergometry (MetaMax 3B-R2, Cortex medical^®^, Leipzig, Germany). The calibration procedure was performed at the laboratory site prior to testing in the morning. It has to be stated that the air humidity and the temperature levels in the testing site varied from the laboratory conditions. Prior to outdoor testing, no further specific calibration was performed. The Metamax system and the mask were placed on the participating subject at the testing site with a maximum delay of 30 min from laboratory calibration to maintain the stability and sensitivity of the instrumentation.

During the testing days, the environmental conditions showed comparable conditions with similar outdoor temperature, dry TR track, and no rain showers. Immediately after reaching the summit of the test site, the individual time for the test track (recorded time in minutes) and one point lactate concentration to determine peak-exercise lactate level (measured in mmol·L^−1^) with a capillary blood analysis from the earlobe were obtained within the first minute post-exercise in each participating athlete (Lactate Scout 4 and EKF Diagnostics, EKF-Germany^®^, Cardiff, Wales, UK). In between the two uphill locomotion test tracks, one hour of recovery was granted for each participating athlete. The athletes were allowed to recover by resting in a sitting or lying position, using recovery techniques such as stretching, and refueling energy by hydration. For each TR athlete, individual CPET performance was analyzed by breath-by-breath analysis for this outdoor sport-specific field testing. As we were able to define a plateau in our TR athlete’s cardiorespiratory response and taking additional secondary criteria for a maximal effort into consideration [38], we identified the following TR athletes’ maximal cardiorespiratory responses during CPET: individual $\dot{V}O_{2}maximum$, individual $\dot{V}Emaximum$, time to reach anaerobic threshold, maximal breath frequency (BF_max_), peak O_2 pulse_, athlete’s peak-exercise lactate level, and specific Cr data (Cr locomotion _mean_). The time of uphill locomotion was also acquired and analyzed. The ventilatory thresholds VT1 and VT2 during the outdoor field testing were determined as described before in our previous research on outdoor uphill testing in TR athletes [22]. According to di Prampero et al., 1986, and Vernillo et al., 2017, “the energy cost of running (C_r_), is defined as the amount of energy spent to transport the subject’s body a given distance” [12,15]. In our study, we calculated the Cr based on the formula described by Balducci et al. [1]:$$Cr_{\;}=\frac{\left({\mathrm{VO}}_{2}\;\mathrm{peak}-0.083\right)}{m\times v};\;\mathrm{with}\;v\;(m/s{-}^{1}),\;m\;(\mathrm{kg}),\;\mathrm{and}\;\mathrm{the}\;\mathrm{absolute}\;\mathrm{term}\;0.083\;(\mathrm{mL}/O_{2}/s{-}^{1}).$$

Furthermore, the individual TR athlete’s time to reach the respiratory exchange ratio equal to 1.01 (RER = 1.01) was measured, pointing out the maximal lactate steady state (MLSS) during the race performance (Time_RER1.01_ (min)) [39,40].

### 2.5. Statistical Analysis

Data were analyzed with SPSS software version 21.0 (IBM, SPSS^®^ software, Ehningen, Germany). Firstly, all acquired data were assessed for normal distribution by analyzing the data by means of Shapiro–Wilk testing and, secondly, the homogeneity of variances was asserted by Levene’s testing, which showed that equal variances could be assumed.

Afterwards, a *t*-test for paired samples was used as the statistical test for hypothesis testing and to compare the means of the two samples. Therefore, the means of the two samples were compared to determine whether the two samples were different from one another. In calculating the *t*-test, the following three fundamental data points were essential: values including the difference between mean values from each data set, the number of data values, and the standard deviation of each group. Results are presented as mean ± standard deviation. *p* ≤ 0.05 was accepted as statistically significant. Afterwards, a gender-specific analysis for the interesting parameters was utilized equally.

## 3. Results

As described in the methods section, all athletes performed the TR testing under similar environmental conditions. During the testing days, which were conducted over the summer months, the outdoor air temperature ranged from 12 to 25 °C and the air humidity was 50 ± 10%.

The cardiorespiratory and metabolic performance parameters of the outdoor uphill performance testing in our TR athletes are presented in Table 3.

No significant differences could be revealed in the CPET data analyses for the $\;\dot{V}O_{2}maximum$ (mL·kg^−1^·min^−1^) between both analyzed uphill locomotion strategies in the TR athletes (*p* = 0.362, data presented in Table 3). Furthermore, no significant differences were elucidated for the peak O_2 pulse_ (mL/bpm) in the TR athletes (*p* = 0.154, presented in Table 3) nor for the maximum breath frequency during exercise (*p* = 0.191, results presented in Table 3). Additionally, Time_RER1.01_ (min) did not significantly differ in between the two analyzed TR motion performances (*p* = 0.212, data presented in Table 3). The only significant difference for CPET variables between the two locomotion strategies was in $\dot{V}Emaximum$ (L·min^−1^) (*p* = 0.033, results presented in Table 3 and Figure 1A). By analyzing the individual TR athletes’ time to reach the peak finishing line in the uphill outdoor test track (recorded time in minutes), significant differences could be proven, whereby the TR athletes were significantly faster performing the running test motion (Time _uphill_, *p* = 0.009, data presented in Table 3 and Figure 1B).

The gender-specific subgroup analysis, as stated in the limitation section, might point out interesting gender-specific insights in the presented CPET parameters. In this context, no significant (ns) differences between both uphill locomotion strategies were revealed, except the significantly faster uphill running time in male TR athletes (results shown in Table 3).

## 4. Discussion

This feasibility outdoor uphill study was undertaken to determine physiological differences in TR athletes regarding different uphill locomotion strategies, uphill running versus uphill walking. The aim of the presented work was to provide new insights into sport-specific cardiorespiratory demands due to uphill locomotion strategy.

Previous research revealed correlations between performance in endurance running and anthropometric characteristics, such as body fat percentage, body mass, height, and BMI [41,42,43]. The most able runners were shorter and lighter than the other competing athletes [41]. Next to these conditions, evaluating the performance of TR athletes seems to include additional multifaceted aspects, especially environmental conditions, such as topographic uphill running profiles and sport-specific demands on the muscle composition, especially the lower limbs [24].

To date, research has mainly focused on CPET parameters and lactate thresholds concepts to predict an athlete’s performance in road running [44,45], but little is known about performance prediction and LT in TR athletes [46]. As key physiological parameters characterizing an athlete’s running performance, $\dot{V}O_{2}\mathrm{peak}$, $\dot{V}O_{2}$ at lactate threshold, and running economy are known [22,47,48,49,50], whereby the determining physiological factors in uphill running are mainly metabolic, biochemical—such as factors of energy cost—and cardiovascular [33]. Taking these multifaceted aspects into consideration, we were not able to provide significant different cardiorespiratory performance parameters in our TR athletes due to uphill locomotion strategy, except $\dot{V}Emaximum$. These results emphasize the importance of having a good running economy to provide good race performance by the ability to maintain a high intensity level for as long as possible in addition to having a high $\dot{V}O_{2}$_max_ [51]. In our previous research, we could elucidate comparable results in CPET parameters for TR athletes compared to road running athletes with regards to $\dot{V}O_{2}\mathrm{peak}$, $\dot{V}O_{2}$ at LT, and peak O_2 pulse_ [22]. Considering these parameters, the question arises whether there are various multifaceted aspects for uphill TR which might influence an athlete’s performance to be more effective and less energy demanding next to the physiological demands.

In our TR athletes, who seemed to be on a comparable fitness level and who are famous for their enhanced aerobic and anaerobic capacity in uphill locomotion [22,52], we solely could elucidate significant differences due to cardiorespiratory performance parameters for $\dot{V}Emaximum$. Our findings might be influenced by various notable performance parameters: firstly, previous research showed that $\dot{V}O_{2}\mathrm{peak}$ is not estimated to be a systematically reliable predictor of running performance and generally a low variability of $\dot{V}O_{2}\mathrm{peak}$ in these trained TR athletes is observed [2,49]. However, next to $\dot{V}O_{2}\mathrm{peak}$, the term “velocity at $\dot{V}O_{2}maximum$” was introduced in 1984 to combine $\dot{V}O_{2}maximum$ and economy to identify aerobic differences between runners [53]. Secondly, the likely uphill locomotion velocity in our participating athletes on the short TR course might result in our comparable CPET measurements. Ortiz et al., revealed previously that oxygen consumption and metabolic running power (indexed W/kg) increased linearly with velocity in vertical kilometer (VK) race athletes, whereby at speeds slower than 0.7 m·s^−1^, walking required less metabolic power than running and, at speeds of 0.8 m·s^−1^, there were no metabolic cost differences, suggesting that running likely costs less energy than walking in a laboratory setting [54]. Taken together, slower athletes in VK races should walk uphill and faster racers should run to minimize their specific metabolic locomotion power needs and to optimize energetic savings [54]. These findings are supported by Giovanelli et al., who revealed a range of optimal inclines (steeper than 15.8°) during uphill walking and running to reduce energy expenditure [55]. Additionally, the vertical ascent rate might be maximized up to slopes between 15° and 25° by using poles in uphill locomotion to delay fatigue effects [35].

No significant differences could be proven for the parameter lactate peak exercise due to uphill locomotion strategy. Blood lactate levels are known to be influenced by age, sex, training status, and the athlete’s overall effort [9], and previous research revealed higher blood lactate concentrations in uphill running than in level running with a certain response in metabolic variables to increasing slopes [15,22,56]. In this context, experienced runners on uphill grades are able to provide a certain running economy reflecting both intrinsic physiological demands and skill [14]. Additionally, Lemire et al., could reveal that experienced endurance athletes showed a variable cardiorespiratory response to uphill and downhill running with regard to maximal oxygen uptake, heart rate, and ventilation response, and TR athletes did not reach$\dot{\;V}O_{2}maximum\;$during maximal incremental downhill testing [11]. Therefore, multifaceted variables, such as different interindividual athlete’s running economy or higher recruitment of muscle mass during uphill running, might result in an individual athlete’s lactate levels and RER response [9,14,22,57]. An individual selective recruitment of type II glycolytic and type I oxidative muscle fiber activation during uphill locomotion and a variable glycogen depletion during uphill running, especially in the gastrocnemius, soleus, and vastus lateralis muscle—revealed in previous research in humans and rats—entail a certain individual variability in lactate levels and might finally result in comparable results in our athletes [13,58,59]. Referring to these variables, metabolic sport-specific physiological adaptions might result in probably compensatory higher $\dot{V}Emaximum\;$during uphill running locomotion in our TR athletes to compensate higher CO_2_ production in the involved muscles and to buffer the accumulating acid [22].

Another aspect to be focused is that previous research revealed an association between reductions in thoraco-abdominal coordination during uphill running and reduced breathing efficiency with a less efficient ventilatory pattern [60]. Both conditions are known for determining ventilatory efficiency and represent a key tool for an athlete’s performance evaluation [60]. The reduction in thoraco-abdominal coordination during increased slope running displayed by greater forward inclination of the trunk and displacement of the center of mass might influence the obtained changes in ventilatory patterns in our uphill running athletes. The decreased efficient ventilatory pattern represented by an increased respiratory rate for the same amount of ventilation, such as higher $\dot{V}Emaximum$, during uphill running locomotion might be the cardiophysiological consequence. Similar findings regarding a higher VE/VT ratio in previous research, which ratio was not analyzed in our study, indicate the same cardiophysiological adaption [60].

There were no significant differences due to uphill locomotion strategy for athletes’ peak heart rate and peak O_2 pulse_. The previously studied muscle mechanoreceptor responding to muscle stretch with its inhibition of cardiac vagal activity, and subsequent increased heart rate response and enhanced cardiac output, might contribute to our athlete’s comparable exhausting CPET data [61]. Nevertheless, analyzing our obtained results, we were not able to elucidate significant differences in cardiac output due to uphill locomotion strategy in our TR athletes. Nonetheless, trained athletes in general are predisposed for functional and structural cardiac remodeling and enhanced cardiac output during exercise, as described before [62]. Our observed comparable dynamic data might be influenced by the following facts: firstly, the dynamic cardiac output data remain unchanged due to the locomotion strategy because of short TR race duration; secondly, the CPET data display equally exhausting performance parameters because TR athletes provide excellent oxygen extraction during uphill locomotion [22].

In our study, we could not prove significant differences in Cr locomotion _mean_ between different uphill locomotion in TR athletes. Previous research in this scientific area indicated that uphill running in well-trained athletes—with greater maximal power of the lower limbs—showed small changes in running mechanics and subsequently lower fatigue-induced alterations in Cr [2]. Minetti et al., revealed that for both walking and running strategy at a given steep incline—up to 24.2%—metabolic power linearly increases with treadmill velocity [26]. In our TR athletes, the Cr locomotion _mean_ did not significantly differ in between the two uphill locomotion patterns, whereby uphill running locomotion is characterized by a higher step frequency, shorter swing/aerial phase duration, increased mechanical work due to the off-road ground, and a progressive adoption of a mid-to-forefoot strike pattern [2,15]. These effects on muscle contraction pattern and on biomechanics, as well as increased working demands during uphill running locomotion, particularly the hip, might contribute to the varying physiological response as we obtained a significantly higher peak$\;\dot{V}E$ during uphill running in comparison to uphill walking strategy [2,15]. The Cr locomotion _mean_ did not differ in between our athletes’ performance, whereby this fact is confirmed by previous research that showed that greater energetic demands in uphill locomotion were not compensated by lower metabolic demands in downhill running [63].

In summary, different uphill locomotion strategies in TR seem not to be associated with variable physiological CPET parameters—except $\dot{V}Emaximum$. Our findings are supported by previous research, whereby $\dot{V}O_{2}$ at a specified velocity and rate of $\dot{V}O_{2}\mathrm{peak}$ did not allow to predict an athlete’s vertical race performance based on classical physiological parameters [55]. Other predictive variables seem to be more appropriate to better describe TR athletes’ running performance, such as the specificity of the running course profile and individual mechanics of running [2,64,65,66,67]. In the end, it has to be stated that predicting TR race performance due to different uphill locomotion modalities is difficult, as the short TR race performance is related to multifaceted variables. In this context, prolonged concentric and eccentric muscle actions during uphill and downhill sections, the individual athlete’s physiological and biomechanical determinants as well as the topographic trail’s characteristics play an important role [2,15,68].

Our feasibility study is not without limitations. First of all, the sample size of analyzed TR athletes is relatively small, which is due to the fact that a selective local recruitment was performed. This resulted in a heterogeneous distribution of male and female TR athletes with a certain mixture of young and experienced athletes, entailing an interindividual variability in relation to lifetime training hours and training schedule variability, race experience, and age. These characteristics might contribute to a certain standard deviation in our data assessment. Due to the relatively small number of enrolled TR athletes, we are not able to draw statistically reliable conclusions, but we might point out interesting trends in TR athletes’ physiological responses due to uphill locomotion in our feasibility trial. Although the number of participating athletes is small, we did perform sex-specific data analyses to provide novel insights in this scientific field, but we are aware that we would not be able derive statistically reliable conclusions. Secondly, our data assessment was performed in variable athletes’ training periods and under “real world” weather conditions. These conditions might contribute to a certain data variability. Due to the small number of TR athletes, we did not run a control arm for this feasibility study. Additionally, as a limitation, it might be stated that one hour resting and recreational time in between two uphill races might be not enough considering the physiological consequences of a maximal intensity eccentric muscle action exercise. This fact has to be taken into consideration for the design of further studies. Nevertheless, the uniqueness of our feasibility study described characteristic physiological responses in short uphill TR due to locomotion pattern. Furthermore, the topographical profile of the Wiesenttal mountain with its short length and single-incline distance solely represents a testing area for short TR, but does not allow conclusions for long-distance TR. Although our new insights might be regarded as an interesting descriptive feasibility preliminary report, they might open the door for investigating this locomotion strategy deeply with more TR athletes including additive locomotion characteristics, such as step frequency and breathing pattern.

## 5. Conclusions

In conclusion, the findings of our feasibility study on sport-specific physiological energetic demands in TR athletes provide new evidence that significant differences in short outdoor uphill CPET assessment can be identified for different uphill locomotion strategies. For short TR performance, significantly higher $\dot{V}Emaximum$ (L•min^−1^) and faster race performance time during the uphill running strategy could be elucidated. However, this preliminary reporting study indicates that short TR physiological race performance cannot successfully be explained by the classical physiological model of endurance running. Due to multifaceted TR performance variables, our findings provide new insights and might contribute to a comprehensive understanding for individual TR athletes’ race performance and uphill locomotion strategy. Further scientific research might be warranted in order to identify the physiological predictors of short TR performance with a homogeneous group of trained TR athletes and to strengthen the scientific evidence of our reported novel insights in outdoor real-world TR performance assessment.

## Figures and Tables

**Figure 1 life-12-02070-f001:**
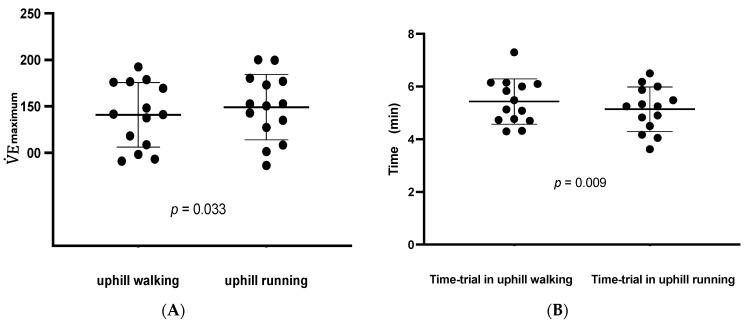
Panel (**A**) Differences in$\;\dot{V}Emaximum$ in uphill walking versus uphill running. Panel (**B**) Time trials in uphill walking vs. running.

**Table 1 life-12-02070-t001:** Anthropometric data of trail running athletes.

	Male n = 10	Female n = 4	*p*-Value
Age (y)	36.2 ± 9.2	38.3 ± 4.0	n.s.
Height (cm)	183.3 + 5.4	168.3 ± 2.5	0.0002
Body mass (kg)	75.3 ± 7.1	58.3 ± 2.2	0.0006
BMI (kg/m^2^)	22.5 ± 1.8	20.8 ± 1.3	n.s.

Data are presented as a mean with standard deviation; Abbreviations: n, number; y, years, cm, centimeter; kg, kilogram; m^2^, square-meter, n.s., not significant.

**Table 2 life-12-02070-t002:** Training and Race Performance Parameter of participating Tral Rummers (n = 14).

Parameter	Meant ± SD
Favorite TR race distance	43.69 ± 26.56 km
Race participation per annum	5.38± 4.41
Current training period	Race: 7.69% Tapering 23.08 Recreation: 69.23
Denivelation running per training week	1200.00 ± 769.58 m
Training distance per week	60.41 ± 26.15 km
Competition in road level running	Yes: 69.24% No: 30.76
Best time in 10 km official race	44.6 ± 0.10 min
Best time in 1000 m denivelation official race	63.83 ± 3.00 min
Years of specific TR training	4.27 ± 3.99
Uphill locomotion strategy - Preferring uphill running - Preferring uphill Walking - Both combined	Running 57.14% Walking 35.71 Both combined: 7.15
Severe Injury break during TR career	Yes: 21.43% No: 78.57

Abbreviations: n, number of athletes; standard deviation; TR, trail running; % percentage.

**Table 3 life-12-02070-t003:** Cardiorespiratory and metabolic performance parameters of uphill TR exercise testing.

Parameter	Uphill Running (n = 14) Male Female	Uphill Walking (n = 14) Male Female	*p*-Value Male	*p*-Value Female	Overall *p*-Value
$\;\dot{V}O_{2}maximum$ (mL·kg^−1^·min^−1^)	57.61 ± 37.0 49.3 ± 3.4 **55.2 ± 7.2**	56.5 ± 6.6 49.0 ± 2.2 **54.4 ± 6.6**	ns	ns	**0.362**
**Peak O**_**2 pulse**_ (mL/bpm)	26.3 ± 2.4 17.5 ± 1.7 **23.7 ± 4.7**	24.8 ± 2.0 17.3 ± 1.0 **22.6 ± 4.0**	ns	ns	**0.154**
**Breath frequency**_**peak**_ (Hz)	55.5 ± 7.7 54.5 ± 8.6 **55.2 ± 7.6**	54.9 ± 7.4 49.0 ± 3.5 **53.2 ± 7.0**	ns	ns	**0.191**
**Time**_**RER1.01**_ (min)	2.0 ± 1.6 3.2 ± 2.8 **1.83 + 1.44**	2.5 ± 2.3 3.3 ± 3.2 **2.73 ± 2.47**	ns	ns	**0.212**
**VE maximum** (L·min)	166.4 ± 23.0 105.9 ± 17.0 **149.2 + 35.2**	158.0 ± 23.7 97.9 ± 7.8 **140.8 ± 34.6**	0.096	0.2330	**0.033 ***
**Peak heart rate** (bpm)	176.2 ± 12.4 171.8 ± 3.0 **175 ± 11**	177.5 ± 11.7 172.3 ± 2.6 **176 ± 10**	ns	ns	**0.297**
**Lacdate**_**peak exercise**_ (mmol·L^−1^)	9.4 ± 3.8 9.7 ± 3.2 **8.7 ± 4.1**	9.6 ± 3.3 8.0 ± 2.1 **9.1 ± 4.3**	ns	ns	**0.752**
**Cr locomotion**_**mean**_ (J·kg^−1^·m^−1^)	7.1 ± 2.0 6.5 ± 2.1 **6.87 ± 2.25**	7.2 ± 2.3 6.6 ± 2.2 **7.02 ± 2.51**	ns	ns	**0.581**
**Time**_**uphill**_ (min)	4.7 ± 1.1 6.1 ± 0.4 **5.13 ± 0.86**	5.1 ± 0.6 6.4 ± 0.5 **5.42 ± 0.86**	0.0217	0.312	**0.009 ***

Data are presented as mean with standard deviation. *p* value *, statistically significant (*p* < 0.05). Abbreviations: SD, standard deviation; Cr locomotion, cost of locomotion; ns, not significant.

## Data Availability

Individual anonymized data supporting the analyses of this study contained in this manuscript will be made available upon reasonable written request from researchers whose proposed use of data for a specific purpose has been approved.

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
