# Peer review of "The Energetic Costs of Uphill Locomotion in Trail Running: Physiological Consequences Due to Uphill Locomotion Pattern—A Feasibility Study"

_life, 2022, doi:10.3390/life12122070_

Round 1

Reviewer 1 Report

General comment: An uphill section of 375 m is very (too?) short.

Abstract: the first sentence is not appropriated and is partly a repetition of line 27.

l.24: remove “in the participating athlete”.

The abstract should be improved to be catchier.

l. 39: this ref is not the best one here. Write “International”.

l. 45: a ref is missing here: https://doi.org/10.1080/02640414.2020.1847502

l. 48: I suggest adding another ref here: DOI: 10.1249/MSS.0000000000002240

l. 55: it is confusing to associate maximal oxygen uptake to the abbreviation of peak oxygen uptake.

l. 58-61: Balducci et al., 2016, Davies et al., 1984, Kasch et al., 1976 and Lemire et al. 2020 are showing that VO2max in level and uphill running are similar, whereas Paavolainen et al., 2000, Scheer et al., 2018, Pringle et al., 2012 and Cassirame et al. 2020 (in steep slopes) are showing the contrary. These 8 studies have to be mentioned in your paper and show the lack of consensus in the scientific literature.

L. 65: considering downhill running, this sentence is not true. Actually, VO2max can not be reached in downhill running (-15% slope) (DOI: 10.1249/MSS.0000000000002240). The value is 16-18% lower than in level or uphill maximal incremental running test.

L. 66-68: Please clarify.

l. 73-75: Ehrström et al. mentioned a third main predictor factor in short trail performance (DOI: 10.1249/MSS.0000000000001467) and also Lemire et al. in uphill running performance (https://doi.org/10.1016/j.jsams.2020.06.004).

l. 83-88: in fact, factors influencing the energy cost of running are rather well identified in uphill vs level running (doi.org/10.1123/ijspp.2021-0047)

l. 95-96: Actually, Giovanelli et al. studied uphill walking vs uphill running and with poles vs without poles. The results should be provided here, and the study cited.

l. 100: VO2peak was already defined, but differently. It should be defined only once. It is confusing to associate maximal ventilation with VEpeak. Please be consistent throughout the ms.

l. 115-117: remove “10 male athletes” since this information is still provided in the table.

l. 114-117 is a repetition of l. 125-126.

l. 127: “inferior to”?

Table 1: substitute “weight” by “body mass” here and throughout the ms, or change the unit to N for Newtons. Specify “N =” in the caption.

Table 2: line 1: add the unit.

Lines 3 and 6: write the unit only once in the first row.

Line 4: write 1200.00 or 770

Line 7: I understand that the participants were not well-trained runners as you mentioned several times in the ms, but rather recreational runners.

Line 8: do you mean vertical race?

Line 9: add the unit in the first row.

Remove “Data are presented as mean with SD”, since it is already written in the title of the second row.

Please specify “N = “.

Km and min belong to the international unit system and do not need to be defined.

l. 140: 29.3% what is the SD value?

What does mean (100 m) here?

l. 159: do not begin a sentence with an abbreviation.

l. 167: did the participants really performed a downhill running time-trial between the two uphill running/walking tests? b[La]- has been measured after the first uphill race and after the downhill race? I am very surprised for different reasons.

Mainly, one could assess that 1 h rest between the 2 uphill races is not enough considering the physiological consequences of a maximal intensity eccentric muscle action exercise.

Why performing a DR time-trial which generates DOMS associated with the eccentric exercise modalitiy.

l. 173: is it peak or max Bf?

l. 173: how did the authors not measure Cr, instead of calculating it indirectly. Cr is provided at submaximal intensity (under VT2) and at steady-state of VO2.

l. 180-190: did the authors test the homogeneity of the data?

l. 186: substitute are by were

l. 193-198: please remove this paragraph, because it is mainly a repetition and add height and body mass (not weight as written) in the method section.

Table 3: The title seems not appropriated, since all data are not reliable to “energetic parameters”

could you provide mean values of VO2, HR, VE and Bf

Line 3: is it mean or peak value of Bf? Change the unit to Hz or breaths/min

Line 6: write HR values without decimals.

Line 9: describe precisely the method for assessing Cr that you used.

Line 11: this time-trial is really short.

Line 12: I still nor understand why the participants did a downhill running time-trial, nor how you use/analyse this result.

Note: please specify the “N = “

The abbreviation of cost of running is Cr and not CR. Cr is not cost of energy (what does that mean?), but cost of locomotion here. All the following abbreviation definitions are useless: it is the international unit system.

l. 203 a dot is missing between kg-1 and min-1.

Figures 1 and 2: remove the titles. The Y axis should not begin at zero. Center the p value. Remove the 2 dashes on the x axis. Present the legend of the x axis in the same order in the 2 figures and rename them “uphill walking” and “uphill running”.

Figure 1 caption: delete the unit and do not write “TR uphill locomotion” because TR excludes walking

Figure 2 caption: I suggest to write “time-trials in uphill walking vs running” for both captions and to merge Figure 1 and 2 in the same Figure with panels A and B.

Figure 2: title of the y axis: write “(min)” instead of “in min”

l. 214-230: these paragraphs are a repetition of the results already presented in the table. Please remove these sentences.

Discussion:

Please apply the admitted method for a discussion section. Remove the lines 234-238 which are close to a ms introduction.

l. 243: not weight but body mass.

Why do the authors discuss correlations, while it was not performed in this study? Did the authors performed correlations between the 2 time-trial performances and the anthropometric values?

l. 252: see again 10.1123/ijspp.2021-0047

l. 252-258: it is not clear which result is discussed here. Please state first a result and then discuss it regarding the scientific literature.

l. 262: based on the table 2, the fitness level of the participants seems to be very common.

l. 265-267: but the velocity at VO2max

l. 270 remove uptake after VO2. It is redundant.

VO2peak here is different to VO2max assessed during a maximal incremental running test.

l. 271: what does mean metabolic power for the author? A power is expressed in W or in W/kg. Did these variables have been investigated?

Specify with or without poles and specify the slope, since the walking/running transition velocity depends on the slope of the terrain (see Nicola Giovanelli’s publications).

l. 276: see my comment about metabolic power above.

l. 279: this b[La]- value is measured after DR TT actually?

l. 281: this statement needs to be nuanced (See Lemire’s publications in MSSE 2020 and JSAMS 2020).

Author Response

Dear Reviewer, please find attached a point to point response.

Thank you very much for your support.

Reviewer 2 Report

Global Comment.

This article provides interesting data and a new approach in Trail Running uphill locomotion. However, you must clarify some points and provide some data to better explain uphill running and walking locomotion.

Methodological section must be clarified, and some results must be included. With these changes you could consider a modulation of discussion

With appropriate corrections this study will provide interesting new insight on locomotion strategy to the literature

Specific comments

Line 2. Change “Trial” by “Trail”

Line 105. There is no hypothesis to this investigation, what is your hypothesis?

Line 107 -132.  You have 10 Men and 4 women. Statistics about both sexes must be separated to provide more relevant insight of subjects. Did you have any idea of VO2 max to judge the real level of athlete.

Line 130. Did you measure a kinetic or only 1 point after uphill. If you measure only one point what time after the arrival of trailers did you measure it?

Line 148 - 178. Could you explain how do you calculate VO2 peak? You also must clarify what methodology you used to calculate anaerobic threshold and Cr.

Line 189 - 191.   You must discuss this element but it’s not a methodological issue. 

Line 193 – 195 Anthropometric data must appear in methodological section

Line 200.Table 3

Why peak O2 is included?

Why did you present mean velocity and downhill time? The topic of your paper in uphill

Why did you only present peak values during uphill? Mean values or kinetics of VO2, VE, BF could help to better understand adaptations during uphill locomotion.

Figure 1 and Figure 2. In figure 1 you have uphill walk on the left and in Figure 2 you have uphill run on the left.

Line 306 – 314. To explain VE data, it could be appreciable to have tidal volume data. During uphill walking runners often “close” their chest hat could alter mechanical of ventilation.

Line 323 – 330. To my mind it’s out of the thematic of this article

Line 343. It’s not the good interline

Author Response

Dear reviewer,

please find attached a point to pint response.

Thank you very much for your support.

Reviewer 3 Report

General comment

The manuscript entitled “ The energetic costs of uphill locomotion in Trial Running: 2 physiological consequences due to uphill locomotion pattern – 3 a feasibility study” aimed to compare cardiorespiratory responses between uphill walking versus uphill running. The article is well written and only minor typing errors needs to be corrected. However, the manuscript really suffer from several inconsistencies, lack of analysis of the literature or results and conclusion not supported by the current findings that preclude its acceptance for publication. The methodological approach needs to be particularly strengthen to be validated, as under its present form too many information are lacking. You will find below my specific comments whereby I encourage the authors to deeply revise their manuscript to consider its publication.

Specific comment

Title: Typing error “Trail Running”

Abstract:

L22: my thought is that the CPET refers to the protocol (i.e. methodological approach & parameters) implemented to assess the physiological demand of uphill locomotion, using a mobile tool (i.e. gas analyzer). I think that authors should reconsider the use of “mobile” CPET to distinguish between the protocol vs the tool used.

L27-28: redundancy with the first sentence of the abstract, please remove the number of males and females athletes here.

L 28: please precise some parameters quantified in the present study (i.e. VE further presented) for CPET parameters and energetic demands.

L 30-31: the second part of this sentence seems to indicate that some athletes displayed greater VE and were fasters than other athletes in your study. But the aim of your study was to consider differences between athletes, but rather differences between uphill walking and running. Please amend or correct this sentence or the objectives for consistency.

Introduction

L 45-46: please precise in which terms trail running can be considered as a “complex sport”.

L 49-51: First, please provide the references for these works. In addition, could the authors define clearly the context of these physiological adaptations? Are they observed in response to acute exercise after a training period? Are they observed after a race, and if so, please place within the framework of the training status and influence of performance levels (e.g. are the best athletes those who experiences greater levels of hyponatremia and muscle damage due to greater intensity during the race?)? The coherence of this sentence with the previous paragraph and the rest of this part is unclear and blur the meaning of this paragraph.

L 51-53: why avoiding here peripheral elements of neuromuscular fatigue?

L 58 and further: what did the authors refer to “cardiophysiological” performance or demand? Is it different from cardiorespiratory function named previously? If not please ensure consistent naming to avoid confusions. Alternatively, please defined what refers to cardiophysiological.

L 59-61: given the amount of literature provided by the authors, and others studies not cited (e.g. Giovanelli et al., 2015, JAP, DOI: 10.1152/japplphysiol.00546.2015; Hoogkamer et al., 2014, PeerJ, DOI: 10.7717/peerj.482; Padulo et al., 2013, Plos One, DOI: 10.1371/journal.pone.0069006), how can the authors state that? Please provide a careful reading about the cited studies, and the available literature.

L 63-65: this sentence is misleading. Balducci and colleagues stated that uphill vs level did not elicited different VO2 max in trained mountain runners. Please consider revising your sentence to avoid misinterpretation.

L 72-73: why the occurrence here of body fat percentage and athletes’ age to explain

L 74: typing error “threshold”

L 81: please specify what parameters of architecture you consider here (i.e. muscle architecture, limbs length?).

L 115-116: participants characteristics should be deleted here and presented in the specific following paragraph.

L 126-127: I guess authors specified minimal and maximal age as well as weight as eligibility criteria rather than predetermined means? Please rephrase to distinguish between these criteria and the resulting characteristics of the included population.

L 129-131: How could 14 athletes been included in the final results if, from the 14 included (eligible?) participants, 2 drop-out during the study? Please report carefully the number of eligible athletes recruited (and included) in the study, and the final number of athletes considered in the results.

L 140: what information refers to “(100 meters)” here? The horizontal distance? If yes, this information could be removed to avoid misinterpretation about topographic characteristics. Authors should rather precise the mean and maximal slopes that could allow experiences readers to better represent the condition of the testing.

L 146: which time window was considered for time to infection here? What about the consideration of musculoskeletal injuries?

L 151: please precise whether the warm-up was freely chosen by participants, or driven by experimenters? How much time separate the end of the warm-up and the beginning of testing?

M 152-153: did the breakfast was controlled? How long between this breakfast and testing?

L 157: please provide the exact model of HR sensor and its validation studies.

L 159-161: please provide how, when and where (testing site or laboratory) the calibration procedure was performed. Additionally, please report the environmental condition during testing.

L 162-165: when precisely was collected the capillary blood sample? Please provide the mean duration between exercise’s ending and lactate measurement. If it is the 1-min post exercise indicated line 168, please consider reorganization of the sentences to simplify reading.

L 165-166: please justify the use of downhill running to return at the start point considering athletes training level and neuromuscular responses to downhill running.

L 169-170: please precise the conditions of the recovery (e.g. environment, position or activity allowed, alimentation and hydration, use of particular gears, …).

L 170: specify upon which analysis stands your measurements (i.e. breath-by-breath, time window).

L 172: how was determined anaerobic threshold here?

L 173: please explain more precisely how was calculated the Cr (analyzing procedure, unit expression, whether RER equivalent was applied)?

Results:

Before presenting the physiological responses to the testing, it could be of particular interest to provide environmental conditions (t°, relative humidity for instance), mean duration of testing to provide an overview of the athletes’ performance.

L 216-219: authors did not mention different lactate measurements / analyses between peak and post lactate. Please amend here and in the method section.

L 227-228: I disagreed with the authors here. Table 3 presented similar velocity but different exercise duration. This does not indicate that athletes were faster in running, but that they cover a greater distance in the walking condition. A careful analysis of the data should be done.

Discussion

L236-238: parameters not measured in the present study, should be removed.

L 241: again a new objective here (sport specific TR metabolism). Please be consistent about the outcomes tested in your study. You specifically mentioned cardiorespiratory and cardiophysiological performance previously, which are not metabolic demands.

L 247-249: I don’t understand what is the information that authors want to provide here.

L 254: everyone can run at high percentage of VO2, but more precisely, the performance is determined but the ability to maintain as long as possible this intensity level, please be precise.

L 262-263: you did not evaluate aerobic and anaerobic capacity in your athletes specifically. This is only the population that could be characterize as having these characteristics. Please correct as required.

L 271: first occurrence of VK please avoid abbreviation here.

L 274-279: the conclusive sentence here is absolutely not supported by your findings, and you did not discuss the origin of the difference in VE between the two conditions. Under its present form, this paragraph could be removed. Please provide greater explanations about your study and the measured outcomes.

L 291-294: although perfectible, how could you state differences in metaboreflex between the conditions while lactate levels are not different? Other explanations based on trunk and upper limb involvements should be provided here to draw a fulfilling discussion.

L 298-299: stating that cardiac output remained unchanged in this sentence is akward. No change relative to rest? Start from finish? Walking from running? Please precise and correct.

L 302: prefer the use of locomotion rather than strategies here.

L 317-320: how could you state that without measurements done for level and downhill sections, and no investigation about muscle activation?

Figures

Table 1: please highlight significant differences between genders

Table 2: Rather than altitude, which is refereeing to the environment, I guess authors referred to the denivelation performed by the participants? If yes please correct. What difference was made to discriminate race period from tapering period to ensure the relevancy of the answers from participants? Did the authors considered the resting period after a race in the “race” period? Did the authors told the participants to discriminate between the main objective and secondary objective to discriminate between “race” and “tapering” period? How was measured or considered the best performance on 1000m denivelation? Did the authors defined ranges for slope or length of races? Particular ground surfaces? Official measurements or performance reported by the athletes? On which basis authors ask the athletes to report their preferred locomotor strategy? Please amend to provide readers relevant information.

Table 3: O2 pulse refers to the oxygen delivery relative to heart rate frequency, not O2 flow, please correct. What difference between peak and post lactate? The Cr value seems really low considering an uphill race here. Were the participants ask to perform their better performance? Your results about the mean velocity and time are fallacious. How could the athletes be as fast during walking than running condition while exercise duration is significantly greater in walking condition? In addition, considering the transition speed between walking and running, it is outstanding that athletes demonstrated similar to higher velocities during walking compared to running. Was the use of poles allowed which could explained this result?

Figure 1: it could be relevant to change graphical presentation in order to associate VE value between walking and running for each athletes. Furthermore, given the superposition of the points, it is really surprising to see a significant different at .033 here. Could the authors add the magnitude of statistical power or effect size here?

Author Response

Dear reviewer,

Please find attached a point to point response.

Thank you very much for your support.

Round 2

Reviewer 3 Report

Main comment

I thank the authors for considering my comments and the high quality of the amendments and corrections performed. The quality of the manuscript has been well improved. I have only three minor considerations that I think could further increase the clarity of the manuscript.

Specific comments

L 207: In my opinion the sentence “the participating athletes were encouraged to proceed a warm-up for 10 minutes prior to uphill locomotion testing” lacks of clarity to clearly define whether all the participants were in the same physiological readiness before the testing? Please be more precise about this particular point.

L 220-222: the calibration site (i.e. indoor) was different from the outdoor, with a mean air humidity of 50%. Did the authors check for the difference in humidity and temperature between the laboratory and the testing site? Could the authors precise whether the Metamax (and the face mask) was placed on the subject soon after the calibration procedure in laboratory, or after a longer delay in the testing site?  

L 236: I am not aware about the comments provided by the other reviewers, but I am not convinced by the change made from authors concerning the maximal cardiorespiratory responses (i.e. VO2, BF, VE, etc). Specifically, unless the authors could observe a plateau in cardiorespiratory responses that would allow them to named ‘maximal response’ (please mention it whenever through), , the VO2, VE or BF values recorded during the uphill section should rather represent peak values. In the absence of a validation procedure it is difficult to interpret whether the participant reached their maximal cardiorespiratory capacities (e.g. see the review provided by Gustavo Z. Schaun, 2017, Sports Medicine, DOI: 810.1186/s40798-017-0112-1).

Author Response

Dear Reviewer,

we highly appreciate the willingness to review our manuscript and also express our thanks for the comments and positive feedback for the authors. Please find below a point-to-point response to the specific comments.
